# The Creative Drummer: An EEG-Based Pilot Study on the Correlates of Emotions and Creative Drum Playing

**DOI:** 10.3390/brainsci13010088

**Published:** 2023-01-02

**Authors:** Rafael Ramirez-Melendez, Xavier Reija

**Affiliations:** Music and Machine Learning Lab, Department of Information and Communication Technologies, Pompeu Fabra University, 08018 Barcelona, Spain

**Keywords:** music, emotion, brain activity, EEG, creativity, arousal and valence

## Abstract

It is reasonable to assume that emotional processes are involved in creative tasks and the generation of creative ideas. In this pilot study, we investigate the emotional correlates in professional drummers during different degrees of creative music playing. Ten participants performed three tasks: repetitive rhythmic drum playing, pattern-based improvisation, and attention-intensive free improvisation, while their EEG activity was recorded. Arousal and valence levels were estimated from the EEG data at baseline and for the three tasks. Results show significantly increased levels of valence (i.e., increased prefrontal right alpha power compared to prefrontal left alpha power) during pattern-based and free improvisation relative to baseline, and significantly increased levels of valence during free improvisation relative to pattern-based improvisation. These results seem to indicate that positive emotion (characterized as increased valence) is associated with the creation of original ideas in drum playing and that the freer the creative process, the greater the positive effect. The implication of these results may be of particular relevance in the fields of music-based therapeutic interventions and music pedagogy.

## 1. Introduction

Creativity has played a central role throughout history. Progress and innovation have been driven by our ability to change existing patterns, and create new things. Given the importance of creative thinking, creativity has become an increasingly popular research topic addressed by various scientific disciplines with a variety of perspectives and methodologies [1,2]. The emergence of new measuring techniques has stimulated research on creativity from different perspectives. For instance, creativity has been studied in the cognitive sciences [3], in pedagogy and education [4], and more recently in neuroscience, (e.g., [5,6,7,8,9]). With a variety of approaches and neuroimaging techniques (e.g., fMRI, PET, NIRS, EEG), neuroscientific studies have discovered correlates between brain activity and underlying creative thinking (for an overview see [8]). For instance, brain-creativity correlates have been investigated in response to divergent thinking [10,11], during insightful problem solving [12] during creativity tasks such as the alternate or unusual uses test [13,14], during match problem solving tasks [15], and during music [16] or visual art [17] imagery and creation.

Neuroscientific research of creativity using Electroencephalography (EEG) techniques has observed alpha band changes compared to control tasks when participants work on divergent thinking tests evaluating the ability to generate multiple solutions to open-ended problems [18,19]. However, there is no consensus among researchers about the direction of such changes. Some investigators have reported frontal alpha increases in synchrony associated with divergent thinking [20,21,22,23,24], as well as increased alpha power at frontal sites [14,23,25,26,27,28,29]. However, the work of other researchers has not been able to replicate these results and has reported decreases in frontal alpha [30,31,32], no significant increases in alpha power [33] or quite the opposite increases in theta, delta, and beta power but not alpha power [10,24,30,31,34,35].

Music is one of the most remarkably creative domains of human activity, so it provides an interesting test case for studying creativity. Early studies on musical creativity include the one by Petsche [16], an EEG study involving seven composers, and the one by [36], a PET imaging study using vocalists. With respect to EEG analysis, Rahman and Bhattacharya [37] perform fine-grain source localization, Petsche [16] focuses strictly on coherence, and Dikaya and Skirtach [38] consider both coherence and power. Regarding the type of participants some studies considered both musicians and non-musicians [39,40,41], while others compared professional and amateur musicians [39], or music students [36], addressing professional and student composers [16,43] or professional freestyle rap artists [42]. Others compared professional pianists with varying degrees of classical vs. jazz training [43,44,45,46] or musicians with more vs. less creative flexibility [47]. Methodologies have been similarly diverse. Bashwiner et al. [39] and Oikkonen et al. [41] used questionnaires to gather information. Studies by Petsche [16] Dikaya and Skirtach [38], and Lu et al. [41] employed mental composition tasks, while others employed a form of improvisation. While composition tasks involved only mental creation (i.e., no motor execution), the improvisation tasks additionally involved motor execution.

There have been several studies investigating music improvisation, both unaccompanied improvisation, and improvisation accompanied by a recording [36,43,45], or another musician [44]. Some studies restricted the set of pitches or rhythms to be used in the improvisation [40,46,48,49,50,51], while in others participants had no restrictions other than the music genre [45], or a particular emotion [51,52]. However, in most studies, the ecological validity was somewhat compromised. For instance, in most piano-improvisation studies, only the right hand was used, which is not representative of real-world performances and affects lateralization and connectivity. Furthermore, with the exception of Rahman and Bhattacharya [37], (which used a full-sized piano keyboard), all piano studies used smaller keyboards of 35 keys [44,45,52], 12 keys [49], 5 keys [40,48], or 1 key [47]. More recent studies related to music improvisation or active playing are based on EEG data [53,54]. Rosen et al. [53] present a study where 32 jazz guitarists improvised to novel chord sequences, while 64-channel EEGs were recorded. Jazz experts rated each improvisation for creativity, technical proficiency, and aesthetic appeal. Their results support a dual-process model of creativity in which experience influences the balance between executive and associative processes. In Sasaki et al., [54] 14 high-level improvisation guitar players played 32-s alternating blocks of improvisation and scales on the guitar while EEG data was recorded. An analysis of the EEG data suggested that improvisation was mediated by processes involved in coordinating planned sequences of movement modulated in response to monitoring and feedback of sensory states in relation to internal plans and goals.

There have been few studies investigating the neural correlates of intentional emotion transfer by music performers. Ghodousi et al. [55] assessed brain activity patterns from the EEG data while participants were performing with emotional intent. The authors contrasted emotional playing with neutral playing to detect patterns of motor and sensory activation related to the emotional aspects of the performance. Pousson et al. [56] recorded EEG activity from musicians who were instructed to perform a simple piano score in an emotional way and in a neutral way. In the emotional playing task, participants were instructed to improvise variations in a manner by which the targeted emotion was communicated, while in the neutral playing task, participants were asked to play the same piece precisely as written in the score. Spectral analysis of the signal was applied finding EEG activity differences between the different conditions. In another study, McPherson et al. [52] used fMRI to examine piano improvisation in response to emotional cues. Twelve professional jazz pianists improvised music that they felt represented the emotion expressed in photographs with a positive, negative, or ambiguous emotion. The authors show that activity in prefrontal and other brain networks involved in creativity is highly modulated by emotional context and that emotional intent directly modulated functional connectivity of limbic and paralimbic areas. Their findings suggest that emotion and creativity are tightly linked. However, the brain activity implications of different degrees of musical creative processes in the emotional state of musicians have not been studied. In particular, to the best of our knowledge, there has been no research on the musicians’ emotional state produced by creative improvisation using EEG data.

In this study, we focus on the musicians’ emotional state during a music creativity task, i.e., music improvisation. With this aim, we asked 10 professional drummer participants to perform three tasks: repetitive drum playing, pattern-based improvisation, and attention-intensive free improvisation. We recorded their EEG activity at baseline while they performed these three tasks, and analyzed the EEG data to estimate instantaneous arousal and valence values. Finally, we compared the estimated arousal and valence values of the different conditions.

## 2. Material and Methods

### 2.1. Participants

Ten right-handed professional drummers (all male, mean = 34 years old, SD = 7), participated in the study. Their average number of years of experience, and years of improvising experience was 18.7 (SD = 5.2), and 12.9 (SD = 6.7), respectively. All participants were trained in jazz improvisation (5.2 years of average training), were graduates of music schools and conservatoires, and fluently read music scores. On average, participants reported practicing 13 h a week, of which 47.5% of the time was dedicated to improvising. Table 1 shows the participants’ information. Participants conceded their written consent and procedures were approved by the Conservatoires UK Research Ethics committee on 4 April 2017, following the guidelines of the British Psychological Society.

### 2.2. Materials

#### Data Acquisition and Processing

The Emotiv EPOC EEG system [57] was used for acquiring the participants’ EEG data. It consists of 16 wet saline electrodes, providing 14 EEG channels, and a wireless amplifier. The electrodes’ positions were located at AF3, F7, F3, FC5, T7, P7, O1, O2, P8, T8, FC6, F4, F8, AF4 according to the international 10–20 system (see Figure 1). Reference electrodes were placed at P3 and P4 (just above the subject’s ears). Data were digitized using the built-in 16-bit ADC with 128 Hz sampling frequency per channel and sent to the computer via Bluetooth. The obtained EEG data were filtered using a Butterworth 8–12 Hz filter. Data were segmented in windows of 1 s and hop size of 0.1 s (i.e., 90% overlapping 1-s windows). For each window, data were squared and averaged. Normalized relative alpha power (alpha = alpha/total_power) was computed. The electrode contact impedance to the scalp was visually monitored using the Emotiv Control Panel software v1.0.0.4.

The Emotiv EPOC EEG device is a low-cost EEG device, which has been mainly marketed as a gaming device. It captures a lower-quality signal compared to the quality of the signal captured by more expensive equipment. However, recent reports evaluating the reliability of some low-cost EEG devices, such as the Emotiv Epoc EEG device, for research purposes suggest that they can be reliable for measuring EEG signals [58,59,60]. A usability review of the Emotiv EPOC EEG device as well as of other low-cost systems can be found in [58]. For recording and processing the data, the OpenViBE platform [61] was used.

### 2.3. Methods

Participants were informed about the procedures and objectives of the study, and were asked to sign the informed consent form. Each session was individual and consisted of three randomly ordered tasks: rhythmic exercise, pattern improvisation, and free improvisation, performed in drum pads. All drummers performed on the same drum kit arrangement and same acoustically-isolated room. Table 2 describes the three tasks performed by the participants. Figure 2 shows the rhythmic exercise task performed by the participants.

EEG data were recorded before the session (2-min baseline recording) and during the rhythmic exercise, pattern improvisation, and free improvisation tasks. In addition, participants responded to 3 qualitative questions before and after the sessions. Before the session and after being instructed about the details of the task, participants were asked how easy/difficult they found the task and how well/badly they expected to perform them. After the session, they were asked how easy/difficult they found the task and how well/badly they had performed them, as well as what was the degree of their general satisfaction with the session. Participants responded to the questions on a 1–7 Likert scale where 1 = extremely difficult, 7 = extremely easy for the first 2 questions, and 1 = extremely dissatisfied, 7 = extremely satisfied for the third question.

#### EEG Analysis

Using the participants’ EEG data, a coordinate in Thayer’s arousal-valence emotion plane [62] was estimated (a representation of Thayer’s arousal-valence plane is shown in Figure 3). The EEG data processing was inspired by Ramirez and Vamvakousis [63] where it was shown that the computed arousal and valence values indeed contain meaningful information about the user’s emotional state. Artifact detection/elimination was performed by visual inspection of the signal. EEG data were normalized to avoid inter-participant variability. Using the EEG signal of a participant, the arousal level was computed as the inverse of the alpha (8–12 Hz) brainwaves (see Equation (1)). EEG data was recorded in 4 locations on the prefrontal cortex: AF3, AF4, F3, and F4 (see Figure 1). Alpha (α) waves are associated with relaxed or brain inactivation states of mind. Thus, the 1/α ratio may be considered an indicator of the arousal state of a person. More precisely, the instantaneous arousal level of a participant was computed as specified by Equation (1) below:Arousal = 1/(αF3 + αF4 + αAF3 + αAF4)(1)

A number of EEG studies [64,65,66,67] have shown that the right hemisphere is more involved in negative emotion while the left frontal area is more associated with positive affect and memories. Thus, for computing valence states, similarly to [63] in this study we computed the activation levels of the two cortical hemispheres and compared them. Positions F3 and F4 are the most commonly used positions for looking at this valence-related activity, as they are located in the prefrontal lobe, which plays a central role in emotion regulation. Valence values were obtained by computing the difference of alpha power α in channels F4 and F3. More precisely, the valence level was computed as specified by Equation (2), as follows:Valence = αF4 − αF3(2)

While the 1/alpha ratio is a reasonable and common way to estimate the arousal state of a person, valence computation is motivated by psychophysiological research, which has shown the importance of the difference in activation between the cortical hemispheres. Left frontal inactivation is an indicator of a withdrawal response, which is often linked to negative emotion. On the other hand, right frontal inactivation may be associated with an approach response or positive emotion. As mentioned above, AF3, F3, AF4, and F4 are the most commonly used positions for computing arousal and valence, as they are located in the prefrontal lobe, which plays a central role in emotion regulation. More details about the way arousal and valence levels are computed can be found in [63,68].

## 3. Results

Using the EEG data obtained, average valence and arousal values were computed before and after the sessions as well as during the three music tasks, (Table 3). Average valence values in Table 3 correspond to the average degree of relative alpha activity in the left frontal lobe, thus larger values are associated with more positive emotional states. Average arousal values on the other hand correspond to less alpha activity (or both) in the frontal lobe, and thus larger values represent higher arousal states.

The normality of the data in all conditions was tested with the Shapiro–Wilk test for normality (*p* ≤ 0.05); data did not differ from a normally distributed data set. A test for a within-subjects, repeated measures study design was performed so as to verify whether there was a significant difference through the conditions. A paired-samples ANOVA was run, thus allowing us to contrast the scores for the different conditions.

Table 3 shows the average and standard deviation values for arousal and valence for the baseline, rhythmic exercise, pattern improvisation, and free improvisation tasks. The computed average arousal values (standard deviation) for these four tasks were 0.8 (0.34), 0.69 (0.31), 0.88 (0.26), and 0.81 (0.3), respectively. Computed average valence values (standard deviation) were −0.24 (0.28), −0.21 (0.36), −0.09 (0.35), and 0.48 (0.27), respectively. After the Bonferroni correction, no significant difference was found between the arousal baseline values and those of the other three conditions. Similarly, no significant difference was found between the valence values of the baseline condition and the rhythmic exercise condition. However, a significant difference was found between the valence values of the baseline condition and the pattern improvisation condition (*p* = 0.000017), the baseline condition and the free improvisation condition (*p* = 1.5 × 10^−56^), and the pattern improvisation condition and the free improvisation condition (*p* = 4.9× 10^−44^).

Figure 4 shows the arousal values in box plots for the baseline, rhythmic exercise, pattern improvisation, and free improvisation tasks. Similarly, Figure 5 shows the valence values for the same five conditions.

Table 4 shows the answers (from 1–7) to the qualitative self-assessment the participants responded to before the session and at the end of the session. For the three questions: how easy/difficult they considered the task and how well/badly they expected to perform them, how easy/difficult they found the task and how well/badly they had performed it, and how satisfied they were with their performance, the average (and standard deviation) of their responses was 5.3 (1.25), 6.1 (0.73) and 6.4 (0.84), respectively.

## 4. Discussion

Electroencephalography data obtained showed that overall valence level in the participants was significantly higher during the pattern improvisation and free improvisation tasks compared to the baseline level. In addition, the valence level was significantly higher during free improvisation tasks than during pattern improvisation. No significant difference was found between the valence levels of the rhythmic exercise and the baseline conditions. These results should be interpreted as a decrease of relative alpha activity in the left frontal lobe during improvisation processes, which may be interpreted as an improvement of mood or a lessening of negative mood [64,68,69]. This seems to indicate that music improvisation has a “feel good” effect and that the more creative the improvisation process, the more intense this effect. This can certainly have implications in improvisation-based music interventions where the goal is to improve emotional conditions.

No significant differences in terms of arousal were found between the baseline, rhythmic exercise, pattern improvisation, and free improvisation conditions. These results should be interpreted as a non-significant difference in alpha activity in the frontal lobe during improvisation processes compared to baseline activity. This may be consistent with the lack of consensus about frontal increases of alpha power at frontal sites during creative tasks.

There is a broad field of research investigating how music playing and music listening can improve the health and well-being of individuals. The unique characteristics of music improvisation may have also unique health and well-being effects compared to other musical activities [70]. As a therapeutic intervention, music improvisation has been used to improve many conditions [71,72,73,74] including mental health conditions and to reduce stress and anxiety [75,76,77,78] The results presented in this study support and provide a ground for the empirical application of music improvisation in such therapeutic contexts. Furthermore, although music improvisation in therapeutic interventions has a different purpose than improvisation in other contexts, its processes can be seen as substantively similar, suggesting that music improvisation can offer intrinsic benefits to the health or well-being of broader populations outside the therapeutic context. The results presented in this study may be interpreted in this direction and may open a more widespread practice of music improvisation in educational and other contexts. Music improvisation and its associated emotional effect may be used as a tool to increase motivation in music students which can lead to a lower abandonment rate.

An analysis of the qualitative self-reported data confirms that the tasks requested to the drummer participants were indeed not too complex for them. The pre-session evaluation of the easiness/difficulty of 5.3 out of 7 (1.25) indicates participants felt capable and confident to perform the task as instructed, which was further confirmed by the post-session evaluation of 6.1 out of 7 (0.73). Furthermore, post-session results showed a high degree of satisfaction among participants (6.4 out of 7), which in turn is an indicator that they understood the instructions well and they were able to execute the tasks correctly. This was important since we wanted to make sure that the tasks were not too complex for the participants so they could fully concentrate on the improvisation component of the pattern and free improvisation tasks.

The presented study has several limitations which should be considered in future research. One of these limitations is the small number of participants. Extending the number of participants would very likely provide more reliable and significant results. Future research could also include involving musicians with different improvisation training experience in order to compare the role of training in the results. While most of the music improvisation activity happens in the context of ensemble playing, most studies, including the present study, consider solo improvisation performances. In this respect, it would be interesting to extend the current study considering the more ecologically valid test case of ensemble music performances, where the EEG data are recorded from musicians while they interact with each other.

## 5. Conclusions

We have investigated the emotional correlates in the emotional state of professional drummers during different degrees of creative music playing, i.e., music improvisation. With this aim, ten participants performed three tasks: a repetitive rhythmic exercise, a pattern-based improvisation, and an attention-intensive free improvisation task, while their EEG activity was recorded. Arousal and valence levels were estimated from the EEG data at baseline and for the three tasks. Results showed significantly increased levels of valence (i.e., increased prefrontal right alpha power compared to prefrontal left alpha power) during pattern-based and free improvisation relative to baseline, and significantly increased levels of valence during free improvisation relative to pattern-based improvisation. These results seem to indicate that positive emotion is associated with the creation of original ideas in drum playing and that the freer the creative process, the greater the positive effect. The implication of these results may be of particular relevance in the fields of music-based therapeutic interventions and music pedagogy.

## Figures and Tables

**Figure 1 brainsci-13-00088-f001:**
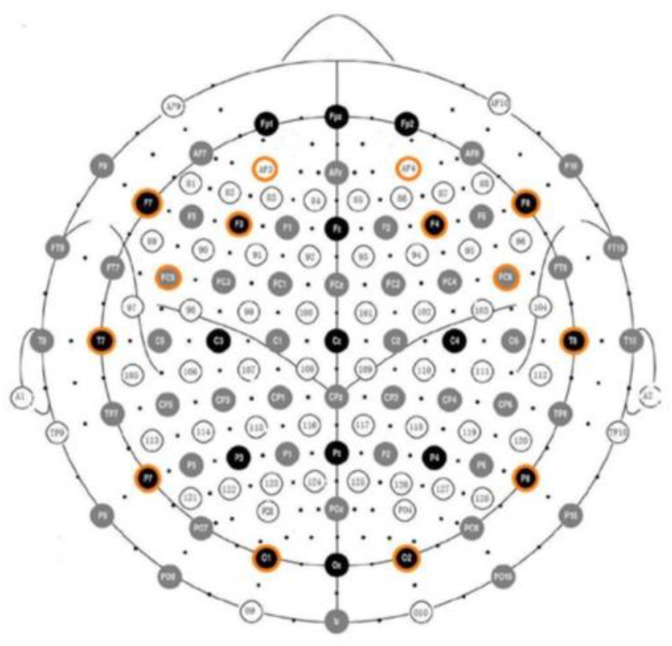
Electrode positions in the Emotiv EPOC (shown in orange) according to the international 10–20 system.

**Figure 2 brainsci-13-00088-f002:**
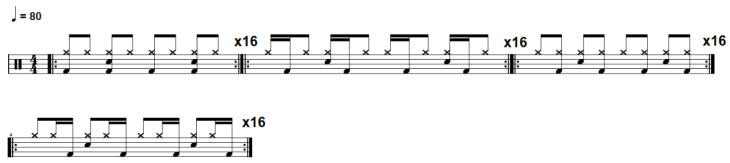
Rhythmic exercise task performed by the participants.

**Figure 3 brainsci-13-00088-f003:**
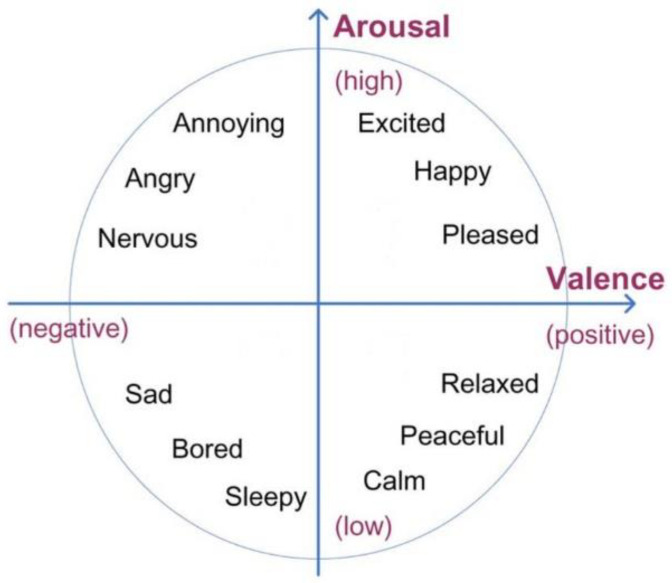
Thayer’s arousal-valence emotional plane.

**Figure 4 brainsci-13-00088-f004:**
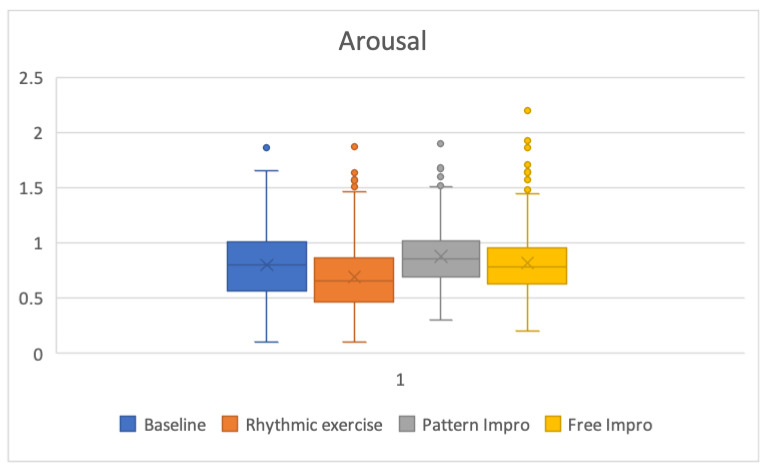
Box plot of the arousal values for the baseline, rhythmic exercise, pattern improvisation, and free improvisation tasks.

**Figure 5 brainsci-13-00088-f005:**
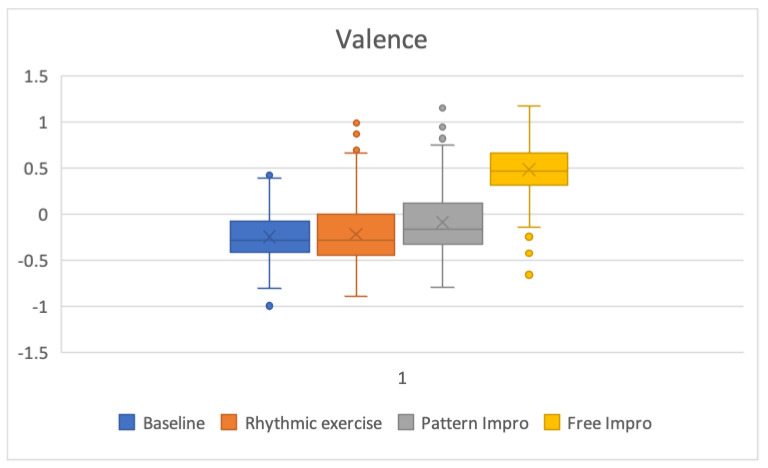
Box plot of the valence values for the baseline, rhythmic exercise, pattern improvisation, and free improvisation tasks.

**Table 1 brainsci-13-00088-t001:** Participants’ drum playing experience (years), improvisation experience (years), practice (hours/week), improvisation (% of the practice), and age.

Participant	Playing Experience (Years)	Improvisation Experience (Years)	Practice (Hours/Week)	Improvisation Practice %	Age
P1	25	15	14	25	40
P2	14	11	8	60	29
P3	20	15	20	30	40
P4	12	2	4	25	34
P5	15	10	8	90	31
P6	15	8	20	25	31
P7	12	5	20	60	23
P8	21	10	10	30	35
P9	25	25	12	40	41
P10	28	28	14	90	46
Avg	18.7	12.9	13	47.5	35
SD	5.2	6.7	6	22.6	6.8

**Table 2 brainsci-13-00088-t002:** Tasks performed by participants.

Task	Instruction	Duration	Eyes
Rhythmic exercise	Perform the same rhythmical pattern repetitively. Participants were instructed to play a 4/4 time-signature pattern executed with both hands, in which the right-hand plays eighth notes, the left-hand plays quarter notes in beats 2 and 4, and the right foot plays the first 4 rhythmic melodies in the sixteenth note subdivision.	5 min	Closed
Pattern improvisation	Improvise concatenating predefined drum patterns (licks). Participants were instructed to develop an improvisation based on the repetition of mechanical patterns, based on muscle memory.	5 min	Closed
Free improvisation	Improvise freely without resourcing to predefined drum patterns. Participants were instructed to develop a free improvisation where the only requirement was to sing what is being played at each moment, no matter, difficulty, or performance, no judgement of good or bad improvisation.	5 min	Closed

**Table 3 brainsci-13-00088-t003:** Average and standard deviation values for arousal and valence for baseline and the rhythmic exercise, pattern improvisation and free improvisation tasks.

Indicators	Baseline		Rhythmic		Pattern Impro		Free Impro	
	Avg	SD	Avg	SD	Avg	SD	Avg	SD
Arousal	0.8	0.34	0.69	0.31	0.88	0.26	0.81	0.3
Valence	−0.24	0.28	−0.21	0.36	−0.09	0.35	0.48	0.27

**Table 4 brainsci-13-00088-t004:** Participants’ qualitative self-evaluation of the session/tasks.

Participant	Easiness of the Task Pre-Session (1–7)	Easiness of the Task Post-Session (1–7)	Degree of Overall Satisfaction (1–7)
P1	7	6	7
P2	4	6	6
P3	4	6	7
P4	4	6	5
P5	4	4	7
P6	6	7	7
P7	6	5	6
P8	6	7	7
P9	5	6	5
P10	7	7	7

## Data Availability

Data is currently unavailable due to privacy restrictions.

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
