# Peer review of "The Creative Drummer: An EEG-Based Pilot Study on the Correlates of Emotions and Creative Drum Playing"

_brainsci, 2023, doi:10.3390/brainsci13010088_

Round 1
Reviewer 1 Report
An excellent and very useful study using electrophysiological markers for creativity/improvisation and emotion in musicians!!!!
I would like to know the years of music learning; are all graduates from musical schools/ or Conservatories?
Are all fluent musical readers; is important to know that information!!
Author Response
Thank you for your comments. The information you mention has now been added to the text (line 121):
Their average number of years of experience, and years of improvising experience was 18.7 (SD=5.2), and 12.9 (SD=6.7), respectively. All participants were trained in Jazz improvisation (5.2 years of average training), were graduates of music schools and conservatoires, and fluently read music scores.
Reviewer 2 Report
In the reviewer's opinion, the manuscript is a rush submission, and the analysis is insufficient to generalize the results. Neither the findings nor the significance of the study are novel or interesting in any way. In addition, the main contribution that the authors have made to the literature review is not very exciting. In addition, the gap that they have identified in the literature is not well defined by the authors. The novelty of the proposed approach is limited at the moment, and several technical details are missing; specific points need to be clarified or improved. Currently, it does not meet the standards of this journal for publication. I doubt the generalizability of these results. I don't think it was well written and had many technical problems. This manuscript is not acceptable to me because of specific aspects of its technicality and presentation.
Author Response
Without more specific comments on the technical and presentation deficiencies of the paper, it is impossible for us to improve the paper or discuss them.
Reviewer 3 Report
The authors presented a study correlating EEG of Emotion states and Creative Drum Playing. As this is an EEG study, I first focused on the methodology, and I found some critical issues that need to be addressed before going into the details.
The first critical issue is that the authors mentioned two frequency bands (alpha + beta) in the preprocessing step, but later only used alpha. I wonder what happened with the beta band? What about other EEG frequency bands?
The second and most important part is related to the extracted features (power bands). There are aspects and formulas of the frequency bands and emotions, but no explanation of how the power was calculated. I found only a single mention of the word "power" in line 194. A clear explanation of how the power spectra (wavelets, welch, on what time segments, length...) were obtained is required for correct discussion of the obtained results.
Furthermore, it is important to state if used absolute power or normalized relative power, (e.g. alpha=alpha/(total power)) was used. Combining multiple subjects with non-normalized power can lead to biased results, as the absolute power can be influenced by different factors that are not directly related to the brain activity (gain of the EEG amplifier, electrodes, skull thickness...). I have noticed that line 178 states "EEG data were normalized to avoid inter-participant variability", but it is not clear if the data is meant the power or something else.
Author Response
Thank you for your insightful comments.
More details about the EEG data processing have been added to the text in lines 139:
Data were segmented in windows of 1 second and hop size of 0.1 seconds (i.e. 90% overlapping 1-sec windows). For each window, data were squared and averaged. Normalised relative alpha power (alpha=alpha/total_power) was computed.
Round 2
Reviewer 2 Report
The authors do not provide a clear definition of the gap they have identified in the reviewed literature, and their contribution is not exciting. There is not enough detail in the "Results" section. The reviewer believes the manuscript is a rushed version of the submission, and the authors need to add more data to support their claims. I can't support publishing this poor-quality manuscript in the journal. As a matter of fact, I believe that the editor should reject this manuscript without any further revisions, in my opinion.
Author Response
Once again, thank you again for your comments. Please find our replies to your comments below:
>The authors do not provide a clear definition of the gap they have identified >in the reviewed literature, and their contribution is not exciting.
We have rewritten the last part of the introduction (see lines 100-127) to make more clear the gap we have found in the literature. Basically, while there have been very few studies investigating the neural correlates of intentional emotion transfer by music performers, there has not been any study, to the best of our knowledge, investigating the implications of different degrees of musical creative processes, i.e. music improvisation, in the emotional state of musicians using EEG data.
>The reviewer believes the manuscript is a rushed version of the submission, >and the authors need to add more data to support their claims.
Indeed as stated at the end of the discussion section (lines 333-335) the small number of participants is a limitation of the study. In order to stress this fact, we have modified the title of the manuscript to "...an EEG-based Pilot Study..." (and abstract) to stress the fact that this is a pilot study, hopefully preceding other studies with more participants.
